# Therapeutic Effects and Underlying Mechanism of SOCS-com Gene-Transfected ADMSCs in Pressure Ulcer Mouse Models

**DOI:** 10.3390/cells12141840

**Published:** 2023-07-13

**Authors:** Youngsic Eom, So Young Eom, Jeonghwa Lee, Saeyeon Hwang, Jihee Won, Hyunsoo Kim, Seok Chung, Hye Joung Kim, Mi-Young Lee

**Affiliations:** 1Department of Medical Science, College of Medical Sciences, Soonchunhyang University, Asan 31538, Republic of Korea; 2School of Mechanical Engineering, Korea University, Seoul 02841, Republic of Koreahyunsookim486@gmail.com (H.K.);; 3Division of Pulmonary, Critical Care and Sleep Medicine, Department of Internal Medicine, Eunpyeong St. Mary’s Hospital, College of Medicine, The Catholic University of Korea, Seoul 34943, Republic of Korea; 4KU-KIST Graduate School of Converging Science and Technology, Korea University, Seoul 02841, Republic of Korea; 5Center for Brain Technology, Brain Science Institute, Korea Institute of Science and Technology (KIST), Seoul 02792, Republic of Korea; 6Institute of Chemical Engineering Convergence System, Korea University, Seoul 02841, Republic of Korea

**Keywords:** adipose-derived mesenchymal stem cell (ADMSC), suppressor of cytokine signaling (SOCS), gene transfection, pressure ulcer (PU)

## Abstract

Although the proportion of ulcer patients with medical problems among the elderly has increased with the extension of human life expectancy, treatment efficiency is drastically low, incurring substantial social costs. MSCs have independent regeneration potential, making them useful in clinical trials of difficult-to-treat diseases. In particular, ADMSCs are promising in the stem cell therapy industry as they can be obtained in vast amounts using non-invasive methods. Furthermore, studies are underway to enhance the regeneration potential of ADMSCs using cytokines, growth factors, and gene delivery to generate highly functional ADMSCs. In this study, key regulators of wound healing, SOCS-1, -3, and -5, were combined to maximize the regenerative potential of ADMSCs in pressure ulcer treatments. After transfecting SOCS-1, -3, -5, and SOCS-com into ADMSCs using a non-viral method, the expression of the inflammatory factors TNF-alpha, INF-gamma, and IL-10 was confirmed. ADMSCs transfected with SOCS-com showed decreased overall expression of inflammatory factors and increased expression of anti-inflammatory factors. Based on these results, we implanted ADMSCs transfected with SOCS-com into a pressure ulcer mouse model to observe their subsequent wound-healing effects. Notably, SOCS-com improved wound closure in ulcers, and reconstruction of the epidermis and dermis was observed. The healing mechanism of ADMSCs transfected with SOCS-com was examined by RNA sequencing. Gene analysis results confirmed that expression changes occurred in genes of key regulators of wound healing, such as chemokines, MMP-1, 9, CSF-2, and IL-33, and that such genetic changes enhanced wound healing in ulcers. Based on these results, we demonstrate the potential of ADMSCs transfected with SOCS-com as an ulcer treatment tool.

## 1. Introduction

Pressure ulcers (PUs) refer to damage to the skin or underlying tissue from ischemic tissue necrosis caused by the obstruction of blood capillaries when pressure or pressure in combination with friction and shearing force is applied continuously and repeatedly to localized areas of the body. Pressure ulcer sores are classified based on their state and severity into conditions such as non-blanchable erythema and partial-thickness tissue loss [1]. With the increase in human life expectancy, the incidence of pressure sores in elderly patients with chronic medical conditions, rather than quadriplegia caused by accidents and diseases, has been increasing. Pressure ulcer sores continue to cause problems, and their extremely low treatment efficacy results in a financial burden on the healthcare system, costing approximately $25 to $96 billion per year in the United States [2].

Mesenchymal stem cells (MSCs) can be extracted from various human tissues, such as bone marrow, fat, synovium, cartilage, skin, and placenta. MSCs can self-renew and differentiate into osteoblasts, adipocytes, and chondrocytes under specific conditions, and they react to biological signals related to inflammation, necrosis, and tissue injury [3,4,5,6]. While the basic therapeutic mechanism of MSCs is not fully understood, it is generally postulated to be the secretion of trophic factors, such as direct differentiation and growth factors, immune factors, antimicrobial peptides, chemokines, and extracellular vesicles. Therefore, MSCs are considered a therapeutic option for injured tissues or disorders without effective treatments [7]. Adipose-derived mesenchymal stem cells (ADMSCs) have become the most popular cell source in cell-based therapy because they can be easily and repeatedly derived through minimally invasive techniques compared to other MSC sources [8]. Numerous animal studies have shown that ADMSC treatment of chronic wounds can lead to improved wound healing and vascularization, skin healing without necrosis or uncontrollable pain, accelerated wound closure, improved epidermal and dermal architecture, and reduced inflammation [9,10,11]. Additionally, studies to verify these effects in clinical practice have been conducted, and the results suggest that ADMSCs improve dermal angiogenesis and remodeling with a high level of complete wound closure [12,13,14]. These clinical and preclinical studies imply that ADMSCs have the potential to be used as therapeutic agents for chronic wounds.

It is very important to improve the limited efficiency of cells to regenerate in order to use MSCs as a healing treatment. Therefore, studies have been conducted by several researchers to improve skin regeneration through gene transfer, which is useful for skin regeneration. A non-viral gene delivery system transferred CXCR4 into MSCs, which improved their homing ability toward the injury site and wound healing [15]. In addition, EGF was gene-transfected into MSCs to confirm interaction with fibroblasts, and increased cell adhesion molecules, actin organization-related proteins, and phospho-(ser) kinase substrates positively affected fibroblast movement and proliferation, which resulted in improved wound healing [16]. However, research on applying such high-efficiency MSCs to ulcer models is insufficient.

Cytokines are involved in paracrine and autocrine cell signaling, are produced or released according to immune events, and are involved in various cellular functions that contribute to differentiation, proliferation, maturation, and wound closure in wound healing. An elaborate diversity of signaling pathways in wound healing is tightly regulated by large amounts of cytokines and growth factors, and these cytokines and growth factors are important mediators for differentiation, proliferation, maturation, and various other cell functions in the wound-healing process [17,18,19,20].

The suppressors of cytokine signaling (SOCS) proteins are negative regulators of cytokine signaling. SOCS proteins are key components of the wound-healing process that regulate the behavior of epithelial cells [19,21]. To date, studies on the role of SOCS proteins in wound healing have been limited and are not well understood. However, several studies have shown that SOCS proteins mediate numerous cytokines and growth factors and play an important role in wound healing. SOCS-1 interacts with IL-2, IL-4, IL-6, INF-γ, and TNF-α to play an important role in fibroblast infiltration, increased fibroblast metabolism, angiogenesis, re-epithelialization, collagen deposition, tissue remodeling, increased vascular permeability, homeostasis, and the provision of metabolic substrates during wound healing [19,22,23,24]. SOCS-3 interacts with IL-1β, IL-2, IL-6, IL-10, INF-γ, TNF-α, EGF, PDGF, HGF, and TGF-β during the wound-healing process and is involved in inflammation, angiogenesis, re-epithelialization, collagen deposition, tissue remodeling, keratinocyte chemotaxis, macrophage and neutrophil migration, and fibroblast proliferation and infiltration [19,25,26,27]. SOCS-5 interacts with IL-4R, IL-6, EGF, and EGFR and is involved in inflammation, angiogenesis, re-epithelialization, collagen deposition, tissue remodeling, induction of fibroblast proliferation, and fibroblast collagen secretion [19,28,29]. Such roles have been explained by several researchers. SOC-1, -3, and -5 play important roles in the wound-healing process. Claire et al. demonstrated that SOCS-1 plays a significant role in inflammatory inhibition [30], and Feng et al. showed that overexpression of SOCS-3 is crucial for wound healing [21,31]. SOCS-5 also acts as a negative regulator of inflammatory cytokines [19]. However, such research results were acquired through the evaluation of a single SOCS, meaning that there were no studies that assessed the effects of the SOCS–cytokine interaction. Therefore, this study aimed to maximize the ulcer wound-healing effect of ADMSCs, which are non-invasive and can be produced in mass amounts by co-transfecting SOCS-1, -3, and -5, which act as key regulators in each stage of wound healing, to enhance the low healing rates of ulcers and confirm their potential as a high-performance ulcer treatment.

## 2. Materials and Methods

### 2.1. Preparation of Cells

ADMSCs were purchased from Lonza (Basal, Switzerland). The cells were cultured in Dulbecco’s modified Eagle’s medium (DMEM; LONZA, BioWhittaker^TM^, Walkersville, MD, USA) supplemented with 10% fetal bovine serum (FBS) and 1% penicillin–streptomycin (PS). The cell culture medium was refreshed every 2 days, and cell detachment for subculture was performed using 0.05% trypsin-EDTA (1×, Phenol Red, Gibco, Waltham, MA, USA).

### 2.2. Cloning of the SOCS-1, -3, -5, and SOCS-com Expression Plasmids for Transfection

The clones were purchased from Addgene (Watertown, MA, USA) to amplify SOCS-1, -3, and -5. Using PCR, each gene was amplified. The amplified genes were applied in the cloning process using the pEGFP-C1 expression vector, either independently as SOCS-1, -3, or -5 or all at the same time as SOCS-com, which combines SOCS-1, -3, and -5. The cloning was performed by BIONICS (Seoul, Republic of Korea).

### 2.3. Electroporation and Confirmation of Transfection Efficiency

For the transfection of ADMSCs with the SOCS-1, -3, -5, or SOCS-com plasmids, the Neon™ Transfection System (Invitrogen™, Neon™, Carlsbad, CA, USA) was used. After harvesting subconfluent MSCs, 1 × 10^6^ cells were resuspended in R buffer, provided by the manufacturer, to which 2 μg of constructed DNA was added. Through repeated tests, the optimal conditions for the transfection of ADMSCs with SOCS-1, -3, -5, or SOCS-com plasmids were set at a voltage of 900 V, a width of 40 ms, and 2 pulses. The gene transfection efficiency was determined by fluorescence microscopy (DMI400B, Leica Microsystems, Heerbrugg, Switzerland, and BD LSR II flow cytometry, BD FACSCanto II, BD Biosciences, Franklin Lakes, NJ, USA) after 48 h of transfection. The efficiency of the gene transfection was confirmed by real-time quantitative PCR.

### 2.4. RNA Isolation from Transfected ADMSCs

The DNA-transfected ADMSCs were cultured for 2 days, detached, and lysed using QIAzol lysis reagent (Qiagen, Hilden, Germany). After lysis, RNA was isolated from each tissue sample using the RNeasy Lipid Tissue kit (Qiagen, Hilden, Germany), according to the manufacturer’s protocol. 

### 2.5. Real-Time Quantitative Polymerase Chain Reaction (RT-PCR)

RNAs isolated from individual DNA samples were synthesized to cDNA using the High-Capacity RNA-to-cDNA™ kit (Applied Biosystems™, Waltham, MA, USA) by following the manufacturer’s protocol. Then, the RNA level was determined by quantitative real-time PCR using LightCycler^®^ 480 SYBR Green I Master (Roche, Swiss, Basel, Switzerland) by following the manufacturer’s protocol. Table 1 lists the primer sequences used for the PCR.

### 2.6. Pressure Ulcer (PU) Mouse Model

All experiments involving animals were approved by Soonchunhyang University’s Institutional Animal Care and Use Committee. (SCH19-0011; 3 February 2020) Mice were anesthetized, and their hair was shaved and cleaned with 70% ethanol. The dorsal skin was gently pulled up and trapped between two round ferrite magnetic plates that had a width of 10 mm, a length of 10 mm, an extent of 100 mm^2^, and a thickness of 2 mm, with an average weight of 1.63 g and 2200 G magnetic forces (NeoMag Co, Ichikawa, Japan). This process creates a compressive pressure of 50 mmHg between the two magnets. It has been demonstrated that an external pressure of 50 mmHg is sufficient to induce skin necrosis and ulcers by reducing the blood flow by 80%. Dorsal skin was trapped between the magnetic plates for 24 h, and the plates were then removed for 12 h. The dorsal skin was then trapped between the magnetic plates for 12 h, and the plates were removed for 12 h. The mice were not anesthetized during ischemia or immobilized. All mice developed two square ulcers separated by a bridge of normal skin. Each wound site was digitally photographed at the indicated time points after wounding [11]. To assess the effects of SOCS-com-transfected ADMSCs on the development of ulcers in the mouse pressure ulcer model, ASCs were incubated under hypoxic conditions (1% O_2_, 5% CO_2_, and 94% air) for 24 h. Consequently, ADMSCs (4 × 10^5^ cells/200 μL HumanTein Essential Matrix; ROKIT Health Care, Seoul, Republic of Korea) or the same volume of HumanTein Essential Matrix (as a control) were injected into the dermis around the pressure ulcer area just after reperfusion (at the day of reperfusion: day 0). The mice were injected with the substance and euthanized with CO_2_ to obtain skin tissues for histological analyses. As soon as the damaged skin turned into a homogeneous brown patch (5 d), we regularly measured the size of the wound using ImageJ (National Institutes of Health, https://imagej.nih.gov/ij/index.html, accessed on 24 February 2022). In digital imaging, the image of a wound is captured and transferred to a computer. We used the ImageJ program tool to trace the edges of the wound in the photograph and a software tool to measure the area of the wound. The statistical significance was assessed using a one-way analysis of variance (ANOVA) and Tukey’s honestly significant difference (HSD) test.

### 2.7. Histological Analysis

The harvested skin specimens were fixed overnight in a 4% paraformaldehyde solution at 4 °C, dehydrated in a graded series of ethanol and xylene solutions, and then embedded in paraffin using a general paraffin-embedding method. The specimens were then cut into sections of approximately 2–3 mm thick and then placed in individualized cassettes and subjected to 13 h of tissue processing (STP120 spin tissue processor, Myr). The sections, which were obtained using a sectioning machine (Finesse ME Microtome, Thermo Shandon, Pittsburgh, PA, USA) to a thickness of approximately 3–4 μm, were attached to the slides. Following deparaffinization and dehydration, hematoxylin and eosin staining was carried out to evaluate cell deformation based on the degree of damage to the epidermis and dermis, collagen fiber deformation, and the infiltration of inflammatory cells. The statistical significance was assessed using the Kruskal–Wallis test.

### 2.8. Western Blot Analysis

In order to analyze the protein expression changes in the ulcer mouse model, the tissue samples were broken using lysis buffer and then sonicated to ensure that the tissue was fully dissolved. The total proteins of the tissues were obtained via centrifugation at 12,000 rpm for 40 min at 4 °C. Sodium dodecyl sulfate-polyacrylamide gel electrophoretic separation of 25 μg of the total proteins was performed, and then the proteins were transferred to polyvinylidene difluoride (PVDF) membranes for 1 h at 400 mA. The PVDF membranes were blocked for 2 h in a blocking solution with 5% bovine serum albumin at room temperature. The membranes were then incubated with the individual primary antibodies diluted in blocking solution (1:1000) for 18 h at 4 °C. The membranes were then incubated with a horseradish peroxidase (HRP)-conjugated secondary antibody (1:5000) for 50 min at 25 °C. The primary antibodies against COX 2, pERK, p-p38, and iNOS and the secondary antibodies against mice and rabbits were purchased from Cell Signaling Technology (Danvers, MA, USA), and the ß-actin antibodies were from Santa Cruz Biotechnology (Dallas, TX, USA). Each membrane was visualized using the Supernova ECL Western blotting detection system (Cyanagen s.r.l., Bologna, Italy), and band images were taken using Sensi-Q 2000 (Lugen Sci Co., Ltd., Bucheon, Republic of Korea). The quantifications of the protein band intensities were performed using the Image J software ver. 1.47.

### 2.9. RNA Sequencing

For mRNA sequencing library preparation, the isolated total RNA underwent processing using the TruSeq Stranded mRNA Sample Preparation kit (Illumina, San Diego, CA, USA), following the manufacturer’s protocol. The resulting libraries were assessed for quality and size using an Agilent 2100 bioanalyzer DNA kit (Agilent, Santa Clara, CA, USA). Library quantification was performed using qPCR on the CFX96 Real-Time System (Bio-Rad, Hercules, CA, USA). Subsequently, the libraries were sequenced on a NextSeq500 sequencer (Illumina) using a paired-end 75 bp plus a single 8 bp index read run.

To compare the expression profiles of the samples, a subset of several hundred differentially expressed genes was selected. Their normalized expression values were subjected to unsupervised clustering using in-house R scripts. Scatter plots depicting gene expression values and volcano plots illustrating expression-fold changes and *p*-values between the two selected samples were also generated using in-house R scripts.

To gain insight into the functional roles of the differentially expressed genes under the compared biological conditions, a gene set overlapping test was performed. This involved analyzing the overlap between the differentially expressed genes and functionally categorized genes, including biological processes from the Gene Ontology, KEGG pathways, and transcription factor-binding target gene sets. The g:Profiler2, version 0.2.0, tool was utilized for this analysis.

## 3. Results

### 3.1. Mechanism of Pressure Ulcers and SOCS-1, -3, -5, and -com Gene Cloning Schematic Diagram

Pressure ulcers usually occur on bony protrusions, such as on the back of the skull, shoulders, hips, elbows, and heels, or on skin in contact with the floor when lying down due to prolonged pressure or shear stress. It partially or completely blocks blood flow to soft tissues. The sites of occurrence and mechanism of such pressure ulcers are schematically shown (Figure 1A). SOCS-1, -3, -5, and -com, which are key regulators of inflammation, proliferation, re-epithelialization, and tissue remodeling, the four stages of wound healing, were inserted into the multicloning site of a pEGFP-N1 vector expressing GFP (Figure 1B).

### 3.2. SOCS-1, -3, -5, and -com Gene Transfection Efficiency and Transfection Effect into ADMSCs

Currently, the treatment of pressure ulcers mainly consists of methods to prevent the various etiologies of pressure ulcers or to help heal wounds. As patients in stages 3 and 4 undergo surgical treatment, along with changes in dressing and pressure distribution, there is a need for a therapeutic agent capable of suppressing or preventing pressure sores from progressing to the next stage. SOCS-1, -3, -5, and -com genes were transfected into ADMSCs to design ADMSCs as highly functional cells that are more effective in treating pressure ulcers in preclinical and clinical trials. To verify the effect and ratio of the SOCS-1, -3, -5, and -com gene transfers into ADMSCs, some were cultured in vitro and analyzed, while others were transplanted into a pressure ulcer mouse model (Figure 2A). When the transfection ratio of SOCS-1, -3, -5, and -com genes into ADMSCs was confirmed using a fluorescence microscope, it was shown that the transfection rate was different among the groups. However, the overall transfection rate remained high (Figure 2B). The gene expression of SOCS-1, -3, and -5 by SOCS-1, -3, -5, and -com genes in ADMSCs was confirmed by real-time PCR. The expression of SOCS-1 increased 200 and 164 times in SOCS-1 and SOCS-com, respectively. SOCS-3, SOCS-3, and SOCS-com showed 221 and 176 times increases, respectively, and SOCS-5, SOCS-5, and SOCS-com showed 195 and 180 times improvements in gene expression, respectively (Figure 2C). Based on these results, we confirmed that gene expression remarkably increased even after transfection with transient genes. TNF-alpha expression was reduced by 3 and 3.86 folds in the SOCS-1 and SOCS-com transfection groups, respectively. INF-gamma expression was reduced by 1.8, 1.3, and 2.7 folds in the SOCS-1, -3, and -com groups, respectively. However, it was confirmed that the expression of IL-10 was improved overall by the gene transfer of SOCS, especially in SOCS-3 and SOCS-com, and was significantly improved by 4.4 and 5.3 times (Figure 2D). These results implied that co-transfection with SOCS-com reduced the expression of inflammation-related factors, increased inflammation, and maximized the ulcer-healing effects of ADMSCs.

### 3.3. Establishment of Pressure Ulcer Mouse Model and Evaluation of Therapeutic Effects of Highly Functional ADMSCs Transfected with SOCS-1, -3, -5, and -com Genes

Overall, SOCS-com-transfected ADMSCs alleviated inflammatory reactions more effectively than the groups transfected with SOCS-1, -3, and -5 alone. Therefore, the treatment effects of alleviating the symptoms of pressure ulcers in the PU model using highly functional ADMSCs could only be expected in SOCS-com. To evaluate the therapeutic effects of highly functional ADMSCs transfected with SOCS-com, a pressure ulcer mouse model was established using magnets (Figure 3). ADMSCs functionally enhanced via gene transfection were suspended in Matrigel and injected into the pressure ulcer site to assess their therapeutic effects 15 days later. When SOCS-com-transfected ADMSCs were injected into the PU models, the wound almost completely healed after 15 d (Figure 4A). In addition, when the wound size was quantified, the maximum wound area was observed on day 2, after which the wound size began to decrease (Figure 4B,C).

### 3.4. Inflammatory Infiltration and Reconstruction of the Dermis and Epidermis in the Ulcer Model Due to the Introduction of SOCS-com 

To evaluate whether co-transfection with SOCS-com promoted the recovery of the damaged skin structure, histopathological analysis via H&E staining was performed (Figure 5). On day 5, inflammatory cells and loose connective tissue were also observed in the group in which ADMSCs were transfected with the vector, but the extent was not as severe as that in the PU model. The damaged epidermal and dermal layers in the SOCS-com group partially recovered, with some inflammatory cell infiltration in the damaged epithelium. However, the overall inflammatory cell infiltration was significantly reduced in the SOCS-com group, suggesting that the wound was beginning to heal after SOCS-com transfection (Figure 5A).

On day 15, in the PU group, the injured epidermis and dermis regenerated more than those on day 5, and the dermal tissue showed lower connectivity than that in the normal group. The infiltration of inflammatory cells was reduced compared to that on day 5 of the PU treatment. In the vector group, the epidermal and dermal layers regenerated almost perfectly, and the infiltration of inflammatory cells was notably reduced. However, the dermis of the vector group exhibited loose connectivity similar to that of the PU group. In contrast, the dermal tissue exhibited a dense and well-connected structure, and the epidermal and dermal layers were fully restored in the SOCS-com group (Figure 5B). In addition, the overall number of inflammatory cells was dramatically reduced (Figure 5C,D). Taken together, the comprehensive histological results indicated that co-transfection with SOCS-com reduced the widespread inflammation response and accelerated the wound-healing process the most, demonstrating its superior therapeutic potential.

### 3.5. Examination of Protein Expression Involved in Inflammatory and Immune Regulation 

We determined the expression of COX-2, pERK, p-p38, and iNOS, which are involved in inflammatory and anti-immune regulatory factors during wound healing, in tissue samples from a PU mouse model. The expression of COX-2, a pro-inflammatory enzyme, was markedly reduced in the SOCS-com injection group. The expression of pERK and pp38, which are also involved in inflammation, was downregulated in the SOCS-com group. The expression of iNOS, which is involved in the synthesis of inflammatory NO molecules during wound healing, was also markedly diminished in the SOCS-com group (Figure 6A). Protein expression was quantified by measuring band intensity. In the PU model, SOCS-com ADMSCs decreased the expression of COX-2 by 5.4 times, pERK by 3.34 times, p-p38 by 10.5 times, and iNOS by 2.2 times (Figure 6B). These results show that the expression of inflammatory factors is markedly downregulated by SOC-com ADMSCs.

### 3.6. Mechanism of Pressure Ulcer Treatment by SOCS-com-Transfected ADMSCs

RNA sequencing was performed to investigate the therapeutic effects of ADMSCs transfected with SOCS-com, which showed the greatest therapeutic effects among the transfected ADMSCs. Compared to the control ADMSCs, SOCS-com transfection led to changes in the expression patterns of many genes (Figure 7A,B). A total of 108 genes showed high expression levels at Q values less than 0.5, and 66 genes showed low expression levels. The wound-healing process is divided into the following four phases: homeostasis, inflammation, proliferation, and remodeling. Figure 7C shows the roles of the genes overexpressed in SOCS-com-transfected ADMSCs at each step. The differences in gene expression levels are shown in various colors, with red indicating the greatest difference, followed by orange, yellow, and green. Our findings showed overexpression of CXCL1, CXCL5, CXCL8, CCL5, and PLA1A in the homeostasis phase and CXCL1, CXCL5, CXCL8, CCL5, CCL7, TNFRSF8, TNFRSF18, IL32, and CSF2 in the inflammation phase. CXCL1, CXCL3, CXCL5, CXCL6, CXCL8, CCL20, MMP1, ICAM1, and STC1 are involved in the proliferation phase, whereas IL33, MMP1, and MMP9 are overexpressed in the remodeling phase. In the four phases of wound healing, the genes overexpressed by SOCS-com-transfected ADMSCs promoted the transition to the next phase, which led to the pressure ulcer treatment effects of ADMSCs transfected with SOCS-com.

## 4. Discussion

Various studies have evaluated the effects of MSCs on chronic wound healing in animal models and demonstrated that MSCs effectively enhance angiogenesis, wound healing, and re-epithelialization while reducing inflammation [7,32,33]. MSCs improve wound healing by mobilizing various angiogenic factors, including SDF-1, VEGF, EGF, insulin-like growth factor-1 (IGF-1), angiopoietin (Ang)-1, keratinocyte growth factor (KGF), MMP-9, macrophage inflammatory protein (MIP)-1α and β, and erythropoietin (EPO) [34,35]. BMMSCs are considered the most suitable cell source [36], but peripheral blood-derived mesenchymal stem cells (PB-MSCs), human umbilical cord-derived mesenchymal stem cells (hUC-MSCs), and adipose-derived mesenchymal stem cells (ADMSCs) are also being studied in various ways. In particular, ADMSCs are considered a source of attractive cell therapy because they have similar characteristics to BMMSCs, have a short doubling time, and can safely and easily obtain a large amount of cells [37]. Additional clinical studies have investigated the role of MSCs in chronic wound healing, improvement of dermal angiogenesis and remodeling, reduction of ulcer size, and acceleration of re-epithelialization [7,14,38,39]. Although these preclinical and clinical trial studies have shown MSCs’ chronic wound-healing effects, they have limitations in their methodology. Therefore, to heal chronic PU wounds with MSCs, methods must first be developed to enhance and maximize the wound-healing ability of MSCs.

In a dynamic wound-healing environment, SOCS is a key regulator of cytokines and growth factors [20,21]. Several studies have investigated the regulation of SOCS expression and its effects on wound healing. SOCS-1 and SOCS-5 modulate IL-4 signaling to regulate re-epithelialization and the tissue remodeling phase of wound healing [26,30,31], while SOCS-3 plays a role in regulating the inflammatory phase [40]. As SOCS-1, -3, and -5 are important factors involved in different phases of wound healing, the co-transfection of these genes is expected to improve the wound-healing function of MSCs. In this study, we co-transfected ADMSCs with SOCS-1, -3, -5, and -com genes, which were then injected into a pressure ulcer mouse model. In all groups transfected with ADMSCs, chronic wound size and protein expression of iNOS, an inflammatory factor, were reduced. These changes were greater in the SOCS-com group than in the other groups. These findings suggest that the co-transfection of ADMSCs with SOCS-1, -3, and -5 accelerates different phases of chronic wound healing.

Although viral gene delivery is a useful system with high gene delivery efficiency and stable gene expression, it is associated with the risk of immunological reactions and mutagenesis. In contrast, non-viral transfection has low immunogenicity, which allows for simple and safe gene transfection [41]. Therefore, non-viral transfections are commonly used in clinical trials to exclude the risk of viral transfection. In this study, we transfected SOCS-1, -3, -5, and -com into ADMSCs to design highly functional ADMSCs and confirmed in vitro that transfection with SOCS-com is the optimal condition for reducing inflammation and enhancing immune regulation. Based on these in vitro results, we compared the SOCS-com transfection conditions with control conditions in a pressure ulcer (PU) mouse model. The results showed that SOCS-com-transfected ADMSCs enhanced wound closure, downregulated proteins encoding inflammatory and anti-immune regulatory factors, reduced infiltration of inflammatory cells, and stimulated reconstruction of the epidermis and dermis. These results suggest that SOCS-com-transfected ADMSCs are effective in treating pressure ulcers.

In many studies, the mechanism of wound healing has been described as regulated by angiogenic (TGF-β, TNF-α, VEGF, PDGF, FGF, angiogenin, and angiopoietin-1) and anti-angiogenic (angiostatin, TIMP-2, TSP-1, endostatin, sprouty 2, PEDF, PF-4, and IFNα/β) factors, including oxidative stress (HIF-1α), inflammation, and Sonic hedgehog (Shh), which are mainly concerned with the process of angiogenesis [4,42,43,44].

Chemokines are key regulators of the four phases of wound healing, and CCL and other genes are involved in the transition to the next phase [45]. Hemostasis is an essential phase of wound healing that prevents further blood loss at the wound site. The coagulation cascade is immediately activated to form clots. In this phase, CXCL1, 4, 5, 7, 8, and 12 and their ligands, CCL2, 3, and 5, are released and play essential roles in the formation of fibrin clots [45,46]. The recruitment of inflammatory cells helps progress to the next wound-healing phase. In our study, the expression of CXCL1, 5, and 8 and CCL5 was significantly increased by transfection with the SOCS-com gene, suggesting that SOCS genes are involved in the transition from the hemostatic phase to the next inflammatory phase. The inflammatory phase is characterized by an influx of inflammatory cells and proangiogenic molecules into the wound. Chemokines mainly recruit inflammatory cells to eliminate dead cells, debris, and foreign bodies in the wound and promote the differentiation of endothelial cells, endothelial progenitor cells, and keratinocytes to ultimately close the wound. These chemokines, which are expressed during the inflammatory phase, are involved in the recruitment of macrophages and the promotion of angiogenesis. CXCL8 is a strong activator and chemoattractant of neutrophils [47,48]. Upon recruitment to a new wound site by CXCL8, neutrophils release CXCL8 and generate a chemokine gradient with CXCL1, 4, 5, 6, and 7 through interactions with proteoglycans, which recruit leukocytes to the wound site [49]. The increased expression of CXCL 1, 5, and 8 after SOCS-com transfection in our study suggests that SOCS genes contribute to the rapid transition from the inflammatory phase to the next phase. TNFRSF is another large immunoregulatory family member that provides costimulatory signals to many immune effector cells [50,51]. TNFRSF is involved in the inflammatory phase, and CCL5 recruits eosinophils, neutrophils, and monocytes while initiating important events in the inflammatory response [45,52]. CSF2 is actively released by non-hematopoietic cells in response to injury signals and strongly promotes MSC migration and differentiation [53]. The increased expression of TNFRSF8, 18, CCL5, CSF-2, and other chemokines by the transfected SOCS-com gene was expected to have positive effects on the wound-healing process. In the third phase of wound healing, the number of inflammatory cells decreases, followed by re-epithelialization to close the wound. In the early stages of proliferation, various neovessels are present to support rapid cellular proliferation and migration of wounds, which is mediated by pro-angiogenic chemokines, such as CXCL1, 2, 3, 5, 6, 7, and 8 [45]. In the present study, transfection of the SOCS-com gene increased the expression of CXCL1, 2, 3, 5, 6, and 8, thereby inducing prompt cellular proliferation and migration. The SOCS-com gene also increases the expression of MMP-1, a key molecule involved in the regulation of epithelial cell proliferation, by altering the expression of the KGF receptor in wound healing [54]. SOCS-com transfection is thought to overexpress MMP-1 and cytokines involved in the proliferation phase and contribute to the rapid transition to the next phase.

The remodeling phase, the last phase of wound healing, involves wound construction and ECM remodeling mediated by MMP released by macrophages, epidermal cells, endothelial cells, and fibroblasts. IL-33 stimulates re-epithelialization and ECM deposition to accelerate wound healing. The SOCS-com transfection significantly increased MMP-1 and 9 [55,56], as well as IL-33 [57]. This suggests that the transfection of SOCS-com overexpressed chemokines, CCL, and other genes essential for the different phases of wound healing and, subsequently, enhanced the wound-healing function of MSCs.

In conclusion, co-transfection of SOCS-com with SOCS-1, -3, and -5, which are key regulators of the complex and sophisticated processes of wound healing, effectively enhanced the wound-healing function of ADMSCs by increasing the expression of essential chemokines, CCL, and related genes. Our results demonstrate that ADMSCs, which can be easily obtained through non-invasive procedures from different MSCs, transfected with SOCS-com for enhanced function may be a potential treatment regimen for pressure ulcers.

However, to use SOCS-com-transfected ADMSCs for the treatment of pressure ulcers, it is important to mass-produce ADMSCs using uniform transfection. In this study, ADMSCs were transfected on a small scale in our laboratory. Therefore, future studies must seek appropriate strategies to uniformly transfect large-scale ADMSCs with the SOCS-com gene to establish a pressure ulcer treatment strategy.

## 5. Conclusions

In this study, SOCS-1, -3, -5, and -com, which are known to play a positive role in wound healing, were gene-delivered to ADMSCs using a non-viral method to confirm the improvement effect in ulcers. First, the transfection of SOCS-1, -3, -5, and -com genes in vitro showed that the expression of inflammatory factors such as TNF-α and INF-γ decreased and the expression of anti-inflammatory factors such as IL-10 increased. Next, in the in vitro results, the most effective SOCS-com gene was detected using PU mouse models, and ulcers were significantly improved. Finally, RNA sequencing was performed to determine the improvement mechanism of ulcers by SOCS-com gene transfer, and as a result, it was confirmed that the expression of important chemokines involved in wound healing increased. Based on these results, the possibility that MSCs to which the SOCS-com gene was transferred could be used as a useful treatment for ulcers was confirmed.

## Figures and Tables

**Figure 1 cells-12-01840-f001:**
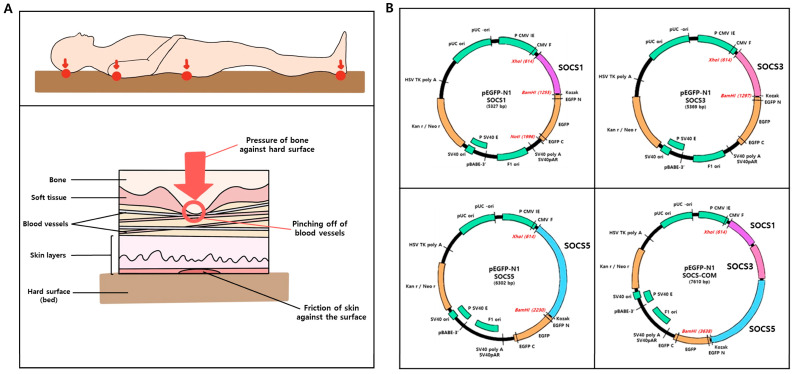
(**A**) Schematic drawing of the cause of pressure ulcers. Prolonged and local pressure, shear, or friction cause pressure ulcers. (**B**) ADMSC plasmid cloning map of each DNA. pEGFP-N1 is used to clone the SOCS cytokine. SOCS-1, SOCS-3, SOCS-5, and SOCS-com are cloned for further experiments.

**Figure 2 cells-12-01840-f002:**
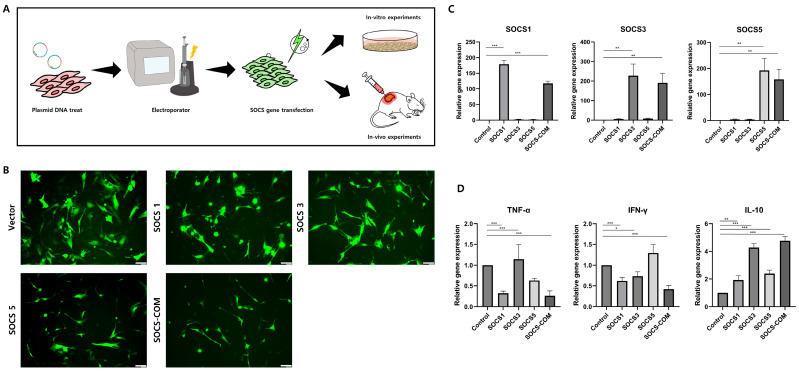
Transfection of SOCS genes into ADMSCs. Schematics showing how a SOCS-com gene is transfected into an MSC using an electroporator (**A**). The transfection efficiency of SOCS-1, -3, -5, and -com plasmid DNAs into ADMSCs was confirmed by fluorescence microscopy (**B**). The transfection efficiency of SOCS-1, -3, -5, and -com into ADMSCs was confirmed by real-time PCR (**C**). Alterations in inflammatory and anti-inflammatory-related gene expressions caused by transfection with SOCS-1, -3, -5, and -com were examined (**D**). Scale bar, 100 µm. (* *p* < 0.05, ** *p* < 0.01, *** *p* < 0.001).

**Figure 3 cells-12-01840-f003:**
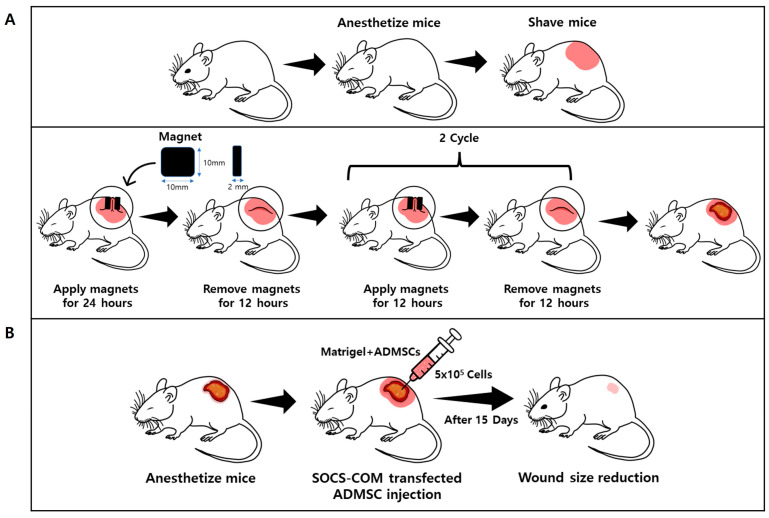
A schematic diagram of experiments for the establishment of a pressure ulcer mouse model and evaluation of the therapeutic effects of highly functional ADMSCs transfected with SOCS-1, -3, -5, and -com. Hair was removed after inducing anesthesia, and magnets were repeatedly attached and removed to induce pressure ulcers. (**A**) ADMSCs transfected with SOCS-1, -3, -5, and -com were injected into the pressure ulcer sites of the mice to evaluate their therapeutic effects (**B**).

**Figure 4 cells-12-01840-f004:**
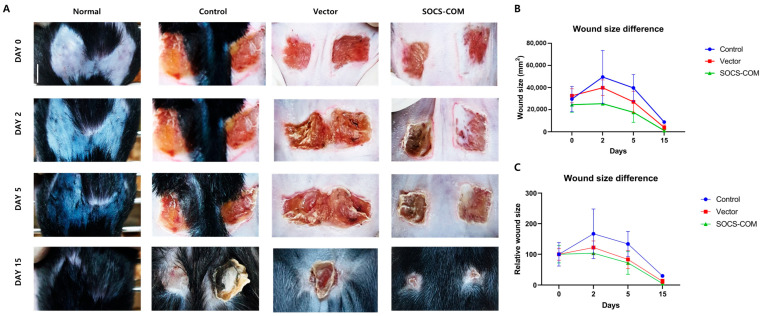
Evaluation of injection effects of ADMSCs transfected with SOCS-com in a pressure ulcer mouse model. (**A**) After injecting ADMSCs transfected with SOCS-com, the therapeutic effects on the pressure ulcers were assessed on days 0, 2, 5, and 15. (**B**,**C**) Wound size differences. The statistical significance was assessed using a one-way analysis of variance (ANOVA).

**Figure 5 cells-12-01840-f005:**
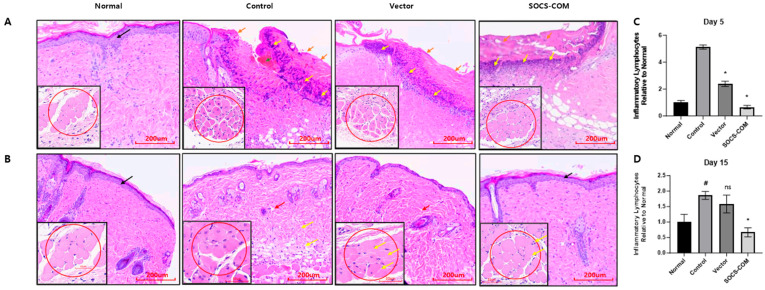
H&E staining and histological analysis of the effect of SOCS-com gene-transfected ADMSCs on the reconstruction of the dermis and epidermis of pressure ulcer mice at days 5 (**A**) and 15 (**B**). Inflammatory infiltration into muscle cells with SOCS-com-transfected ADMSCs in the mouse models was also observed in the bottom box of each figure. Black arrow: intact epithelium; orange arrow: broken epithelium; yellow arrow: aggregation of inflammatory cells, mainly macrophages and polymorphonuclear cells; red arrow: loosely packed connective tissue; green arrow: hematoma. (**C**,**D**) The number of overall inflammatory cells was determined. The statistical significance was assessed using a Kruskal–Wallis test. (* *p* < 0.05 vs. control; # *p* < 0.05 vs. normal; ns *p* > 0.05 vs. control).

**Figure 6 cells-12-01840-f006:**
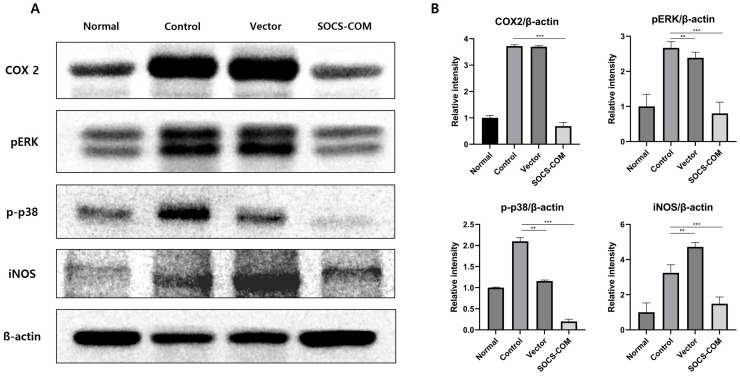
Effect of SOCS gene transfection on the expression of the inflammatory proteins COX 2, pERK, p-p38, and iNOS. Western blot results of COX 2, pERK, p-p38, and iNOS (**A**). Relative expression intensity graph. The expression level of each protein signal was expressed with the relative densities of COX 2, pERK, p-p38, and iNOS to ß-actin (**B**). (** *p* < 0.01, *** *p* < 0.001).

**Figure 7 cells-12-01840-f007:**
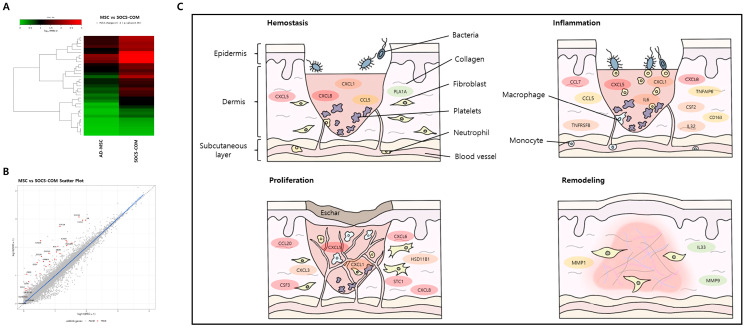
RNA sequencing results of ADMSCs and SOCS-com gene-transfected ADMSCs. (**A**) Heat map showing upregulated genes by transfecting SOCS-com. (**B**) Scatter plot representing major upregulated genes by transfecting SOCS-com into ADMSCs. (**C**) Summarized schematics of wound healing related to overexpressed cytokines by SOCS. The color of the image represents the increased gene-fold rate of SOCS transfection (based on the fold change; green < 6.0; 6.0 ≤ yellow < 8.0; 8.0 ≤ orange < 13.0; 13.0 ≤ red).

**Table 1 cells-12-01840-t001:** PCR primer sequences. RT-PCR primer sequence of each gene.

**SOCS1**	F	CGT GAA GAT GGC CTC GGG AC
	R	GCT CCA GCA GCT CGA AGA GG
**SOCS3**	F	CTA CTG GAG CGC AGT GAC CG
	R	CTG CAG AGA GAA GCT GCC CC
**SOCS5**	F	TGC TCC ATG GGG TGG GAA GA
	R	AGA ACT TAC GCC GTA GCG CC
**TNF-** **α**	F	TTC TCA TTC CTG CTT GTG GC
	R	CTG ATG AGA GGG AGG CCA TT
**IFN-** **γ**	F	CTC TGC ATC GTT TTG GGT TCT CTT GG
	R	GCG ACA GTT CAG CCA TCA CTT GGA T
**IL-10**	F	CCA AGC CTT GTC TGA GAT GA
	R	TGA GGG TCT TCA GGT TCT CC

## Data Availability

Data is unavailable.

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
