# Peer review of "Therapeutic Effects and Underlying Mechanism of SOCS-com Gene-Transfected ADMSCs in Pressure Ulcer Mouse Models"

_cells, 2023, doi:10.3390/cells12141840_

Round 1
Reviewer 1 Report
In this study, the researchers aimed to maximize the ulcer wound healing effect of ADMSC, which are noninvasive and can be produced in mass amounts, by co-transfecting SOCS-1,3,5, which act as key regulators in each stage (inflammation, proliferation, re-epithelialization, and tissue remodeling) of wound healing, to enhance the low healing rates of ulcers and confirm their potential as a high-performance ulcer treatment.
The study is well conducted with necessary controls and recommend the acceptance of manuscript in its current form.
Author Response
Dear Reviewer,
Thank you for your careful review and acceptance of our paper entitled " Therapeutic effect and the underlying mechanism of SOCS-com gene-transfected ADMSCS in pressure ulcer mouse models" We appreciate your time and valuable feedback.
Once again, we thank you for your positive evaluation and acceptance of our paper. We are grateful for your valuable input, which has undoubtedly improved the quality of our work. We look forward to addressing any further comments or queries during the publication process.
Sincerely,
So Young Eom

Reviewer 2 Report
1. Why the normal groups were not included in the Figure 4B-C analysis? Also at day 15 seems like all groups of mice were recovered, what's the benefits of SOCS expression then?
2. Please quantify H&E staining
3. Figure 7 RNA-seq data was not interpreted at all.
Author Response
Dear Reviewer,
We appreciate the valuable comments and suggestions provided. We would like to express our gratitude for taking the time to thoroughly review our work and for your insightful feedback.
Regarding the first comment regarding the figure 4B and 4C, we agree that expressing significance would enhance the robustness of our findings. Actually, the precise measurement of a wound is not easy. Especially, the measurement of wound size is variable depending on wound exudate, the presence of necrotic tissue, slough and granulation tissue, as well as undermining and tunneling.
At day 15, the wound size of SOCS-com-was almost completely closed and well-healed than that of the control and vector (Fig 4A). But the graph (Fig 4B and 4C) could not reflect the notable differences between these groups, as you pointed out. To address this, we have prepared a separate graph specifically depicting the data from the 15th day, highlighting the distinct wound closure trends observed. Furthermore, we acknowledge your observation that the error values decrease on the 15th day, indicating increased precision in the measurements. Although these changes may be less noticeable in the overall graph, they are indeed significant and underscore the effectiveness of the SOCS-COM gene in wound healing.
Regarding the additional analysis of H&E staining, we acknowledge the importance of providing additional quantified statistical tests and ensuring the accuracy of our results. We evaluated our H&E staining result to ensure the validity and reliability of our data analysis. In accordance with your suggestion, the infiltrated inflammatory cells were counted as one of the tools to quantify H&E staining. The result showing the differences in the number of inflammatory cells among groups were added as Fig 5C. The comprehensive histological results indicated that co-transfection with SOCS-com reduced wide-spread inflammation response dramatically and accelerated the wound healing process the greatest, demonstrating its superior therapeutic potential.
Thank you for pointing out the need for additional clarification about figure 7. We will provide more detailed descriptions of the figure 7A and 7B, including RNA sequencing results.
We also appreciate the suggestion to include additional data in introduction and reference to support our conclusions. We will carefully examine our manuscript and consider the most appropriate and informative way to present any additional data, ensuring its relevance to the main findings of our study.
Once again, we sincerely thank you for your thoughtful comments and suggestions. Your input will undoubtedly improve the quality and impact of our research. We will diligently address each point you have raised and strive to make the necessary revisions to strengthen our manuscript.
Best regards,
So Young Eom

Round 2
Reviewer 2 Report
None